

# Hydrography of intertidal environments in Schleswig-Holstein, Germany

Joachim Schönfeld[1], Hermann W. Bange[1], Helmke Hepach[1], Svenja Reents[2]

[1]GEOMAR Helmholtz-Zentrum für Ozeanforschung Kiel, 24148 Kiel, Germany
[2]Wadden Sea Station, Alfred-Wegener-Institut, Helmholtz-Zentrum für Polar- und Meeresforschung, 25992 List/Sylt, Germany

*Correspondence to*: Joachim Schönfeld (jschoenfeld@geomar.de)

**Abstract.** The current status of intertidal waters in the wake of ongoing global change was assessed in a baseline study with a 36 month time series of water level, temperature, and salinity measurements from Bottsand lagoon on the Baltic Sea coast,
and on the mudflats off Schobüll at the North Sea coast of Schleswig-Holstein, Germany. Extreme events, storm surges and heat waves were also recorded in a temporal resolution of 20 minutes. At Bottsand lagoon, the temperatures followed the air temperatures in winter, and were higher than the air temperatures in spring and summer. The annual averages varied from 12.1 to 12.6°C, the air temperatures varied from 11.1 to 11.2°C. The salinities showed one or two months periods of consistently higher or lower values in winter and spring. The annual averages ranged from 14.7 to 16.9 units. The lagoon
showed a different variability than that of the open Baltic surface waters, where the temperatures and salinities were lower in summer and higher in winter. The seasonal salinity differences were less developed in the mid 1960s, when the connectivity of the lagoon with the Baltic Sea was less restricted, and a sandy shoal in the lagoon was not present. In Husum Bight off Schobüll, water temperatures were lower than the air temperatures in winter and higher in spring and summer. The annual average water temperatures ranged from 10.8 to 11.4°C, and the air temperatures from 9.9 to 10.2°C. High waters were
warmer during the day than at night-time in spring and early summer only. The annual average salinities off Schobüll ranged from 24.0 to 27.2 units. The values were higher in summer and lower in winter. This seasonal cycle was related to variations in the Elbe river runoff, which largely influences the salinity in the south-eastern German Bight. The same seasonal cycle was recorded in the Sylt Roads time series. Cross-correlations of the records revealed that it takes seven weeks for an Elbe river freshwater pulse to reach Schobüll, and three weeks more to proceed to Sylt. On average, the salinities were 2.7 units
lower off Schobüll than off Sylt, which mirrors a pervasive gradient of landward decreasing salinities in the Wadden Sea. They were induced by a local, low-salinity lens on top of tidal waters, fed by groundwater seepage or by freshwater runoff. A cross correlation with the precipitation record revealed salinity decreases about one week after high precipitation. The cumulative salt marsh submergence times per period of observation, i.e. inundation frequencies, were very variable at the lower boundaries of the lower and upper salt marsh vegetation zones. The inundation frequencies were consistently higher at
Bottsand than at Schobüll, where the same halophyte assemblages prevailed. As the average salinity was 10 units higher at Schobüll, the differences of inundation frequencies suggest that a certain salinity has to be maintained in the soils to sustain



specific halophyte assemblages. A mass occurrence of small Pacific oyster shells was observed before the vegetation boundary off Schobüll in spring 2024. The data suggested an oyster spatfall triggered by the North Sea heat waves in summer 2023, with temperatures exceeding 23°C, and a subsequent wipe-out during a period of salinities lower than 18 units after an Elbe river discharge event in January 2024. The biotic responses to environmental extremes highlighted the vulnerability of Wadden Sea ecosystems at times of Global Change.

## 1 Introduction

Marginal marine environments, including tidal flats, salt marshes, coastal lagoons and adjoining wetlands are characterised by unique faunal and floral assemblages (Halls, 1997). Most species are widely distributed, breaching bioprovincial boundaries (Lübbers and Schönfeld, 2018), and blooms of invasive species are an increasing threat to indigenous communities (Occhipinti-Ambrogi, 2021). A high tolerance to diurnal and seasonal variations in temperature, precipitation, inundation, pore water content, oxygenation, and chemistry of salt marsh soils, and the salinity of tidal waters was considered as reason for the formation of marginal marine assemblages (Adam, 1990). Despite their tolerance to environmental variability, many species showed a distinct range of distribution in the tidal frame (Scott and Medioli, 1978; Bockelmann et al., 2002; Balke et al., 2011). They commonly inhabit bands displaying the optimum conditions for growth and prevalence (Petersen et al., 2014; Kim et al., 2023). The limits of species' distribution were ascribed to threshold values of inundation frequency, desiccation, freshwater influx, salinity, soil oxygenation, pH and carbonate system parameters of pore waters (Elsey-Quirk et al., 2009; Snedden et al., 2015; van Regteren et al., 2020; Schönfeld and Mendes, 2022). They appeared to be variable among different regions and also depend on the properties of the tidal waters (Balke et al., 2016; Li et al., 2018). The seasonal and inter-annual variability of abiotic environmental parameters is less constrained (Costa et al., 2003), even though this is crucial for an assessment how the assemblages may respond to Global Change (Carrasco et al., 2021; Dowling et al., 2023).

The current climate change in Central Europe has effected an increase in annual mean temperatures of 1.2°C since the beginning of the 20[th] century (Anders et al., 2014), and by a mean sea level rise of 24 cm in the south-eastern North Sea (Dangendorf et al., 2013). Although the annual changes are very small, the effects of seasonal, multi-year variability and extreme events, such as the drought of 2018, on flora and fauna are much more serious (Schuldt et al., 2020). Marginal marine habitats are particularly affected. We observed the immigration of new species, the fragmentation and shifting of distribution areas and changes in the reproductive cycle of many organisms during the last decades. Specific causes and triggering events can rarely be identified or related to instrumental records (Schönfeld, 2018). This is mainly due to the small number and hydrographical stations and their settings.

The thirteen Marine Environmental Measuring Network (MARNET) stations of the Federal Maritime and Hydrographic Agency (BSH) are located far from the North Sea and Baltic coasts. Academic institutes operate three near-shore oceanographic and marine biological time series stations that are operated on a weekly or monthly basis at Helgoland, Sylt,





and Eckernförde Bay. In contrast, the German Weather Service (DWD) operates 22 weather stations in the northern German

State of Schleswig-Holstein recording meteorological parameters every hour. The present investigation was therefore driven by a null hypothesis, i.e. temperatures and salinities of intertidal waters show the same variability than those of surface waters further off shore.

This paper presents the first continuous time series of water level, temperature, and salinity from Bottsand lagoon, located at the Baltic Sea coast, and from the mudflats off Schobüll, located at the North Sea coast, covering three annual cycles from

September 2021 to October 2024. The objectives of our study were (i) to constrain the processes affecting the salinity variability tidal and receiving waters (ii) to decipher the effects of extreme events on water temperatures and salinities, (iii) to assess what has changed since the 1960s, and (iv) to demonstrate by the example of a bloom of the Pacific oyster in 2023, how time series data can be used to develop scenarios of multi-year changes in the species composition and distribution of faunal and floral communities.

**1.1 Geographical and environmental setting**

**1.1.1 Bottsand**

The nature reserve Bottsand includes a coastal lagoon, which is situated north of the village Wendtorf, Plön District, Schleswig-Holstein, Germany (Hammann and Zimmer, 2014). The lagoon is about 0.5 km wide, 1.5 km long, and up to 1 m deep. It is confined to the south by Wendtorf Marina, to the east by a dyke, and to the north by a 1.4 km long beach ridge

with dunes (Schönfeld, 2018). The successive westward progradation of the spit since 1870 has almost completely closed off the lagoon from the Baltic Sea (Schrader, 1990; Knief, 2013). The bottom sediment of the lagoon is fine to medium sand (Grabert, 1971). Bogs developed through aggradation at the northern and eastern margin of the lagoon and later developed into an extensive salt marsh (Wolfram, 1996). They have tapered the northern part of the lagoon to a narrow ditch (Lutze, 1968).

The flora in the lagoon is dominated by macroalgae (*Fucus vesiculosus*, *Zostera marina*), the pondweed *Potamogeton pectinatus*, and sea rush (*Bolboschoenus maritimus*), which is found in near shore areas (Hammann and Zimmer, 2014). The latter is considered as pioneer vegetation. The salt marsh commences with a distinct two step brink of up to 40 cm height in total. A narrow zone with saltmarsh grass (*Puccinellia maritima*) and seaside plantain (*Plantago maritima*) is recognised on the bench of the first step at 0.04 to 0.10 m above German Ordnance level (NN, related to Amsterdam Peil; since 1992 NHN,

related to the geoid surface going through Amsterdam Peil). This strip is recognised as lower salt marsh. A hummocky red fescue lawn (*Festuca rubra*) commences on the top of the second step at 0.32 m NHN. The flora of this upper saltmarsh includes blackgrass (*Juncus gerardii*), salt marsh asters (*Tripolium pannonicum*), brackish reed (*Phragmites australis*), silverweed (*Potentilla anserina*), and prickly saltwort (*Salsola kali*) (Wolfram, 1996). The red fescue lawn is dense up to 0.62 m NHN and passes into a strip of semi-dry flood lawn at 0.75 m NHN, above which quitch grass (*Elymus laxus*),

glaucous bluegrass (Poa humilis), creeping bentgrass (*Agrostis stolonifera*), and catsear (*Hypochaeris radicata*) occur



(Christensen, 2021). Red fescue is recognised up to a height of 0.97 m NHN. A dry grass land follows at approximately 1.0 m NHN. The dry lawn, dunes and beach of the nature reserve is an area of high floral and faunal diversities, and an important breeding and resting place for birds (NABU, 2020).

The hydrography of Bottsand lagoon is characterised by a high variability. Salinities ranging from 8 to 19 units, and
temperatures from -1 to 24°C were reported (Lutze, 1968). The salinity variations are mainly driven by precipitation, minor ground water seeping, and incursions of Baltic surface water during high waters (Schönfeld, 2018). Wind-driven water level changes are an important environmental parameter for the ecosystems at Bottsand lagoon (Baerens et al., 2003). The winds may raise the water level up to 2 to 3 m NHN, and they may fall to 1.5 m below NHN (Sztobryn et al., 2009), whereas the northern part of the lagoon never falls dry (Schönfeld, 2018). The range of astronomical tides is approximately ±0.12 m,
depicting a microtidal regime in the lagoon (Hayes, 1979; Schönfeld, 2018; https://gezeiten.bsh.de/kiel_holtenau, last access: 20th March 2025).

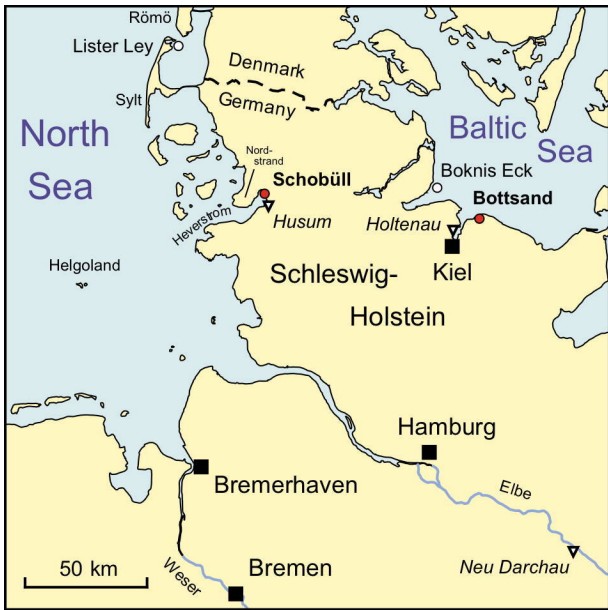

**Figure 1. Overview map of northern Germany with the locations mentioned in this paper. Red dots: intertidal time series stations, white dots: long term monitoring stations, triangles: tide and river gauges. Coastline, rivers and border line adapted from ©**
**Google Maps.**

### 1.1.2 Schobüll

The village of Schobüll is located on the north-eastern side of Husum Bight, a northward embayment of the Heverstrom tidal channel system in the Wadden Sea on the west coast of Schleswig-Holstein, Germany (Fig. 1). The northern part of Husum Bight is characterised by extended mud flats, which are surrounded by salt marshes. They are grazed and managed on the
western and northern side of the bight before the dikes of the isle of Nordstrand and Nordstrand Dam, connecting Nordstrand and the main land. No dyke was built on the eastern side, because the Saalian moraine hill of Schobüll directly adjoins the



Wadden Sea over a distance of 2.9 km. Since the construction of Nordstrand Dam in 1933 to 1935, the hydraulic and sedimentary regime has changed in northern Husum Bight (Stock, 2013). Mud accumulated on near-shore sands, and a salt marsh with a natural succession developed off Schobüll (Lindner, 1952; Lorenzen, 1956). The succession is only biased in

the pioneer vegetation zone and adjoining mud flat, where ditches and groynes for land claims were built and maintained since the 1970s. These measures were reduced since the foundation of the Wadden Sea National Park in 1985, and the salt marsh off Schobüll was abandoned in the 1990s (Stock, 2003). Groynes are only maintained before the vegetation boundary as protection measure. Selected ditches were kept open to facilitate drainage of freshwater springs and wetlands at the foot of the moraine hill.

On the tidal flats off Schobüll, clayey silts accumulate during summer and in calm areas behind groynes. Sandy silts are deposited during winter, and they prevail areas with strong exposure to waves and currents (Unsöld, 1974). In particular, a belt of sandy mud rich in shell debris is developed along the outward groynes and before the vegetation boundary.

The pioneer vegetation zone is characterised by dense meadows of cordgrass (*Spartina anglica*, *Sporobolus anglicus* of authors) commencing at 1.02 to 1.05 m NHN. Saltwort (*Saliconica stricta*) is less common in the pioneer zone at Schobüll

since the introduction of cordgrass in 1927 (Reinke, 1903; Kolumbe, 1931). The lower saltmarsh extends from 1.7 m to 2.2 m NHN and shows patchy occurrences of saltmarsh grass (*Puccinellia maritima*) among a variety of other halophytes, e.g. sea purslane (*Halimione portulacoides*), spear-leaved orache (*Atriplex prostrata*), and salt marsh aster (*Tripolium pannonicum*). The latter may cover up to 20 % of the lower saltmarsh area (Stock, 2013). Cordgrass is also common, in particular in depressions. The upper salt marsh between 2.2 and 2.5 m NHN begins with the first patches of red fescue

(*Festuca rubra*). It continues landwards through dense thickets of brackish reed (*Phragmites australis*). Sea rush (*Bolboschoenus maritimus*) was common on the landward side of the reed fields (Stock, 2013) and is now confined to depressions or areas with groundwater seepage. On elevated areas and on both sides of the pier at Schobüll, a dense, homogenous canopy of sea couch (*Elymus athericus*) has been established, which suppressed other saltmarsh plants. The landward limit of red fescue and wheatgrass marks the highest high-water level around 2.8 m NHN. The salt marsh off

Schobüll provides habitats, food and shelter for many sea bird species, invertebrates, insects, spiders, gastropods, amphipods, and a rich foraminiferal fauna (Lehmann, 2000).

The hydrography of the tidal waters in Husum Bight is sparsely known. Gridded data sets indicate monthly mean surface water temperatures ranging from 3°C in February to 18°C in August, and salinities ranging from 28 to 29 units in winter (December to April) to 29 to 30 units in summer (May to November) for the period of 1900 to 1996, even though the

Wadden Sea is not sufficiently resolved and has a poor data coverage in the northern part (Janssen et al., 1999). Water level data provided a more detailed picture. The mean astronomical tidal range at Husum was 3.51 m, with a mean low water level of -1.81 m and a mean high water level of 1.70 m NHN (Bundesamt für Seeschiffahrt und Hydrographie, 2023). Husum tide gauge is in operation since 1867 (Jensen et al., 1991). Historical tide gauge data analyses revealed a rise in mean sea level of 2.2 mm a$^{-1}$ for the period of 1937 to 2008 (Jensen et al., 2011), while the mean high water levels raised by 3.6 mm a$^{-1}$ on

average from 1934 to 1983 at Husum (Jensen, 1984). The mean high water levels showed an inter-annual variability of





±0.075 m during the period of 2000 to 2022 (Bundesanstalt für Gewässerkunde, 2024), which is effected by the prevailing weather conditions and wind force (Schelling, 1952).

## 2 Material and Methods

### 2.1 Hydrographical measurements

The data for the present study include observations and measurements that were carried out in addition to the annual monitoring of the foraminiferal fauna at Bottsand lagoon (2012-2019) and at Schobüll (2014-2019). A subset of these data from Bottsand lagoon and the methodology has been published by Schönfeld (2018). The hydrographical measurements were resumed in 2021 at the same sites and continued until 2024.

As in earlier studies, Odyssey® data loggers (Dataflow Systems Ltd., Christchurch, New Zealand) were used to measure
water level (P/T), temperature, and salinity (C/T). Xylem Ebro EBI-20T data recorders (Xylem Analytics Sales GmbH & Co. KG, Weilheim, Germany) were used to measure the air temperature at Bottsand and Schobüll. The Odyssey loggers were attached to pressure-impregnated posts or boards with specially designed brackets (utility model DE 20 2020 100 677 U1; German Patent and Trademark Office, 2020). The C/T loggers recording at Bottsand lagoon were mounted to an application-specific holder, a small cylinder with an insert, which was pushed into the bottom sediment. The insert in the
cylinder fixed the data logger in an upright position, in that the device is bathed in lagoonal water and does not get in contact with the ambient sediment. Once the logger is to be replaced, only the insert has to be removed. Two P/T and two C/T loggers were used at each measuring point. Likewise, two EBI 20 temperature loggers were in operation at the air temperature measuring point at Bottsand. All data loggers recorded the current measured value every 20 minutes. They were regularly in operation for four months. At intervals of approximately eight weeks, one P/T, C/T and EBI 20 logger was
replaced at each measuring point while the other continued recording. All loggers were inspected and cleaned from biofilms and barnacles every fourth to sixth week from April to October.

The measuring point at Bottsand lagoon was operated from the 18 September 2021 to 12 October 2024. It was located at 54° 25.610' N, 10° 17.695' E, 1.5 meters off the western bank of the terminal ditch. The assembly consists of two 60 mm diameter posts for attaching the holders and air valves for the P/T loggers. The air valves were mounted to 25 mm
polyvinylchloride tubes with a 1 cm graduation for reading of the pressure sensor immersion depth. Next to the posts, the C/T logger holders were placed in the bottom sediment. The measuring point for the air temperature was established on the edge of the north-western, lower roof beam of the warden's shelter hut, about two metres above ground. The coordinates were 54° 25.517' N and 10° 17.158' E.

The measuring point off Schobüll was operated from the 24 September 2021 to 9 October 2024. It was located at the
seaward end of a groyne, 200 m south of the old pier, at 54° 30.648' N and 8° 59.403' E. The measuring point consisted of a 1 m long board and two fishing rods holding the air valves for the P/T loggers. The fishing rods were inserted into sleeves made of 40 mm polypropylene pipes and fixed with cable ties in order to keep them in a vertical position, even under strong





wind force. Holder for the C/T and P/T loggers and the sleeves were attached to the board and posts on the northern side of the groyne in that they were not exposed to direct sunlight. The holders were fastened with SPAX® INOX screws made of

stainless steel A2, 1.4567, an alloy proven not to be corroded by seawater (Mendes et al., 2025). A supplementary measuring point was installed under the board walk of the old pier at 54° 30.763' N and 8° 59.467' E. Only one C/T logger was installed here with a sensor height of 1.83 m NHN. Immersion of this C/T logger served as independent control of the accuracy of water levels recorded by the P/T loggers. Another supplementary measuring point was installed in the salt marsh at 54° 30.817' N and 8° 59.573' E, where an EBI 20 temperature logger was mounted to a wooden post at 1.50 m above ground and

covered with a white polypropylene beaker. This data logger recorded the immediate air temperature above the salt marsh, which may differ from the temperature at the private meteorological station in the village of Schobüll.

The heights of the data logger holders and sensors were determined after installation by using a Leica Na728 surveyor's level (Supplement Table S1). At Bottsand, the levels were tied to a geodetic reference point at the building Schleusenweg 2 (1.350 m NN; Schönfeld, 2018). At Schobüll, the geodetic point at Schobüll parish church door (12.512 m NN) was used as height

reference. The accuracy of levelling was <±3 to ±5 mm.

The P/T and C/T loggers were recalibrated each time before deployment. A linear regression was used as calibration function to correct the measured values that were recorded with the respective logger during the next four months. The calibration with standard solutions revealed a sensor accuracy of 0.00 to 2.48 cm, on average 0.26 cm for immersion depth, 0.00 to 0.53 K, on average 0.13 K for temperature, and 0.00 to 2.30, on average 0.16 units for salinity (mean of residuals).

The external reproducibility of the P/T and C/T loggers was checked at Bottsand lagoon with regular visual inspections and manual measurements on site by using a WTW LF320 conductimeter with a TetraCon 325 probe. The mean difference of manually measured and logger-recorded values at Bottsand lagoon was 0.03 m for water depth, 0.3 K for temperature, and 0.5 salinity units (n = 26; Supplement Table S2).

For the WTW LF320 device, the manufacturer specifies a precision of <0.5 % for conductivity and <0.1 K for temperature.

Salinities measured with the LF320 deviated on average by 0.1 units, and in two cases by 0.2 units, from the target value of seawater standards with salinities ranging from 4.91 to 34.72 ‰ (Supplement Table S3). The temperatures may deviate by 0.1 K, which is in agreement with the manufacturer's data sheets. For the EBI-20T temperature loggers, a constant correction was applied as provided by the manufacturer's calibration certificate. The accuracy of EBI-20T loggers was given to be ±0.1°C.

**2.2 Data processing**

The water levels were recorded in centimetres above the logger's pressure sensor and converted into meters above NHN using the levelled heights of the data logger sensors. Measurements that indicate an obvious malfunction of the sensor were skipped. From the calibrated and plausibility-checked water level, temperature, and salinity data, mean values were calculated from the measurements of the loggers used in parallel at Bottsand, and for the measurements during high tide off

Schobüll. For the temperatures, only the values recorded by the C/T loggers were averaged off Schobüll, because the devices



were located 27 cm above the P/T loggers. At Bottsand, mean values were calculated from the temperature measurements of all P/T and C/T loggers, because their sensors were approximately at the same height. The difference between the individual temperature, salinity and water level measurements, or the difference between the highest and lowest temperature measurements in Bottsand lagoon, was calculated as the sensor deviation, which depicts the accuracy of individual

measurements on site. For further analyses of the Bottsand data, daily averages were calculated and related to 12:00 CET of the respective day. For the measurements off Schobüll, mean values were calculated from the recordings at high water. The high water times were determined by using the water level measurements from the P/T loggers.

Spring and neap tides were assigned to the high waters according to the lunar calendar. The period of spring tides was defined by the Federal Maritime and Hydrographic Agency as four days from the day of the full or new moon, and the

period of neap tides as four days from the day of the half moon. Instead of the day of the lunar phase, the exact time for 9° E was used. The times and numbering of the new moon phases (lunations) were retrieved from online resources (e.g., https://www.calendar-12.com, https://timeanddate.de). Annual averages for the mean high tide, spring and neap tides were calculated for 12 lunar months between the lunations 1222 (6 October 2021, 12:05 CET) and 1234 (25 September 2022, 22:54 CET), 1235 (25 October 2022 11:48 CET) and 1247 (14 October 2023 18:55 CET), and between lunation 1247 and

1259 (2 October 2024 19:50 CET) (Supplement Tables S4, S5). Storm surges, during which the high tide level is more than 1.5 m above the long-term Mean High Tide at Husum, were also taken into account. Storm surges and individual, significant floods were assigned to the dominating low-pressure areas (https://page.met.fu-berlin.de). The annual mean values of temperature and salinity were calculated over a period of 365 days and included five days before and five or six days after the respective lunar year. The inundation frequency in the salt marshes at Bottsand and Schobüll was defined as percent

submergence time during one lunar year (Li et al., 2018). The submergence times were calculated for levels above the height of the P/T logger sensors by applying the histogram function of Microsoft® Excel® to all water level measurements in the respective lunar year (Schönfeld and Mendes, 2022) (Supplement Table S6).

A cross correlation was applied to constrain the time lag between different data sets. They were brought to the same resolution before the calculation was performed. The program PAST 3.16 was used for cross correlation (Hammer et al.,

240    2001).

## 2.3 Other data sources

Water levels from Holtenau or Husum tide gauges were considered once both P/T loggers at Bottsand lagoon or off Schobüll failed (Fig. 1). The tide gauge data were accessible in real time every minute. They were kept available for 30 days as uncorrected raw values (http://www.pegelonline.wsv.de, last access: 16 August 2024). For the designation of high-water

times and levels at Schobüll, measurements from every 20 minutes were extracted and evaluated.

The hydrographical measurements at Bottsand lagoon were compared to Boknis Eck time series data of surface water salinity and temperature (Bange et al., 2011; Lennartz et al., 2014; Hepach et al., 2024). The Boknis Eck time series station is located at 54° 31' N and 10° 2' E, 19.7 km to the northwest of Bottsand, and 2.14 km off shore (Fig. 1). The station has



been operated on a monthly basis since 1957. The surface water parameters were measured at a standard depth of 0.5 or one
m with an electrical thermometer and salinometer mounted to a CTD, at around 10:00 in the morning. The data from 1957 to
2023 were published in the PANGAEA data base (doi: 10.1594/PANGAEA.855693, 10.1594/PANGAEA.973020).

The measurements at Bottsand lagoon were also compared to previous measurements from December 1964 to May 1967 in
course of the monthly foraminiferal sampling at Station 381 (Lutze , 1968). According to the maps drawn from aerial
photographs, Station 381 was situated at 54° 25.402' N and 10° 17.607'E, i.e. 398 m south of our measuring point in the
terminal ditch of the lagoon. The temperature and salinity data were digitised from Figure 19 in Lutze (1968). The
methodology of hydrographic measurements was not reported.

The measurements off Schobüll were compared to Sylt Roads time series data of surface water and salinity from 2014 to
2024 (de Amorin et al., 2023; Rick et al., 2023). The monitoring station is located in the Lister Ley tidal channel system in
the western part of Sylt-Rømø Bight, Wadden Sea, off the borough of List on the isle of Sylt, Germany, at 55° 1.80' N and 8°
27.60' E. It is about 67 km north of Schobüll, and 1.6 km off the isle (Fig. 1). The station is sampled twice a week since
1973. Sea surface temperatures were measured with a reversing thermometer on site. The salinity of surface water samples
was measured with a Guildline Autosal 8400B salinometer on shore. The data up to May 2019 were published in the
PANGAEA data base (Rick et al., 2023, there Table 5).

Meteorological data used for comparison are daily mean air temperatures at two m height above ground, daily precipitation,
and solar radiation recorded at Schobüll private weather station (https://www.schobuell-wetter.de/monats-jahresdaten-
02.php#a4232, last access: 12th March 2025). Fluvial discharge data from the river Elbe measured at Neu Darchau river
gauge (Measuring point 5930010) were made available for the years 2014 to 2024 by Wasserstraßen- und Schifffahrtsamt
Elbe, Magdeburg, Germany. Weekly mean values were calculated from the daily mean discharge data provided.

## 3 Results

### 3.1 Bottsand lagoon

The 37 months data record from Bottsand was almost complete, no total failures occurred or instruments were lost.
However, both C/T loggers were recovered from beneath the ice on the terminal ditch on 14 December 2022 as precaution
measure to prevent damage or loss of the instruments. They were deployed again on 26 December 2022 after the ice melted.
The temporary removal effected a gap of 1.06 % of the total salinity record. Short periods when the C/T loggers fell dry at



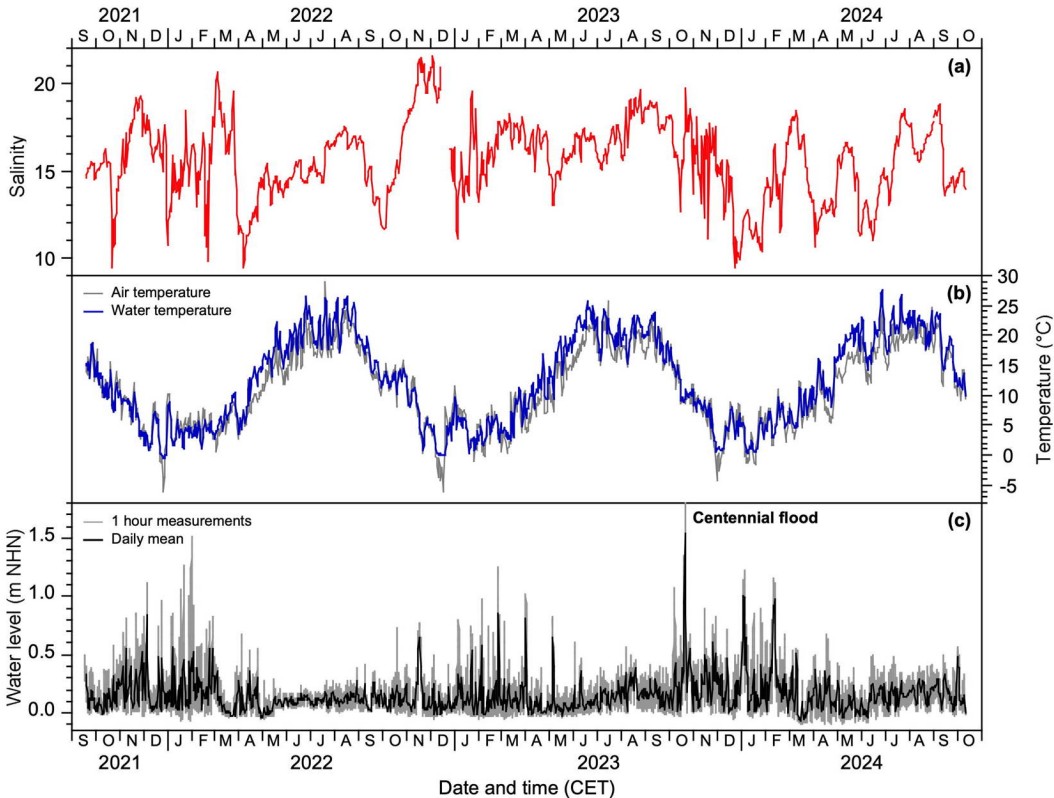

**Figure 2. Daily mean salinity (a), temperature (b), and water level (c) at Bottsand lagoon.**

low water levels resulting in an additional loss of 0.08 % of the salinity record. Failure of single instruments occurred several times due to loose contacts of the conductivity sensor, blocking of the P/T logger air valve or pressure sensor tubing. This also includes the centennial high water on 20 October 2023, when the water level exceeded the height of the air valves at 1.67 m NHN. Consequently, 12.2 % of the water level and 18.6 % of the salinity data were based on the record of a single device. Double measurements revealed a mean sensor deviation of 0.03 m for water level (n = 70861), 0.9 salinity units (n = 64894), 0.8 K for water temperature (n = 80670), and 0.2 K for air temperature (n = 80655).

The water level recorded at Bottsand lagoon ranged from -0.09 to 1.67 m NHN (Fig. 2), with a mean value of 0.15 ± 0.18 m NHN (1 sigma, n = 80673). The mean of a lunar year ranged from 0.13 m NHN in 2022/2023 to 0.19 m NHN in 2023/2024. The record showed a pronounced seasonality. The cold seasons from October to April were characterised by a high variability and frequent high waters culminating in January to February, and attenuating in April to May. Many high waters lasted less than 24 hours and hence are insufficiently displayed by the daily averages. Three to six high waters per season exceeded the upper limit of the semi-dry flood lawn at 1.0 m NHN. The warm season from May to September showed low and tidally-driven water level variations with rather constant daily means (Fig. 2). The daily mean values generally rise during the summer.



The salt marsh submergence times were very variable among the tree lunar years of the investigation period. They ranged from 74.8 to 60.4 % of observation time for the base of the lower salt marsh at 0.04 NHN, from 17.9 to 8.9 % of the time for the base of the upper salt marsh red fescue lawn at 0.32 m NHN, from 0.8 to 2.4 % of the time for the base of the semi-dry flood lawn at 0.75 m NHN, and from 0.2 to 1.1 % for the top of the flood lawn and highest halophyte occurrence at 1.0 m NHN.

The lagoonal temperatures ranged from -0.7°C to 31.8°C, with a mean value of 12.4°C ±7.6 K (1 sigma, n = 80674). The annual mean ranged from 12.1°C in 2021/2022 to 12.6°C in 2023/2024. The temperature fluctuations are small and follow the daily mean values of the air temperature during the winter months. In spring and summer, fluctuations are higher, and the daily means of the water temperatures are usually higher than those of the air temperatures (Fig. 2). A consistent feature of the temperature records are three to eight days periods of moderate to severe air frost in late November to December every year, during which the water temperatures fell to 0.6 to -0.2°C. One or two short intervals of light air frost may occur in January or February as well, during which the water temperatures fell to 0.4°C on just one day, as in mid-January 2024. Ice on the lagoon was observed in December 2022 only. The annual mean air temperatures ranged from 11.1°C ±7.1 K (1 sigma) in 2021/2022 to 11.2°C ±7.3 K (1 sigma) in 2022/2023.

The salinities ranged from 5.4 to 22.3, with a mean value of 15.6 ±2.3 units (1 sigma, n = 79756). The annual mean ranged from 14.7 in 2023/2024 to 16.9 in 2022/2023. The strongest salinity fluctuations were recognised in winter, with generally high values in November and December, and low values in January and February. After high values in March and low values in April to May, the salinities increased again. The summer trend was interrupted by one to two month minima that did not occur at the same time every year (Fig. 2).

### 3.2 Schobüll

The 36.5 months data record from Schobüll was almost complete, 2126 high waters were recorded and 19 (0.9 %) were missed due to water levels of less than 0.67 m NHN. Failures of both P/T loggers occurred during four periods of 10 to 41 days during winter. The fishing rods holding the air valve broke off, topped over, and the air valves were immersed in the mud. Differences between atmospheric and water pressure could not be measured anymore and no water levels were recorded. Tide gauge records from Husum were used to fill these gaps, which comprise 193 high waters (9.1 %). Failure of single instruments or sensors occurred several times due to loose contacts of the conductivity sensor, leaks in the logger casing, battery failures, and aging of electronic components. These failures effected that 376 of 1936 high water levels measured. In addition, 691 of 2091 high water salinity and temperature data were based on the record from a single device. The salinity and temperature of 27 high waters could not be measured because the water level was lower than 0.96 m NHN. Double measurements, when both devices were in operation, revealed a mean sensor deviation of 0.05 m for water level, 0.7 salinity units, 0.4 K for water temperature.





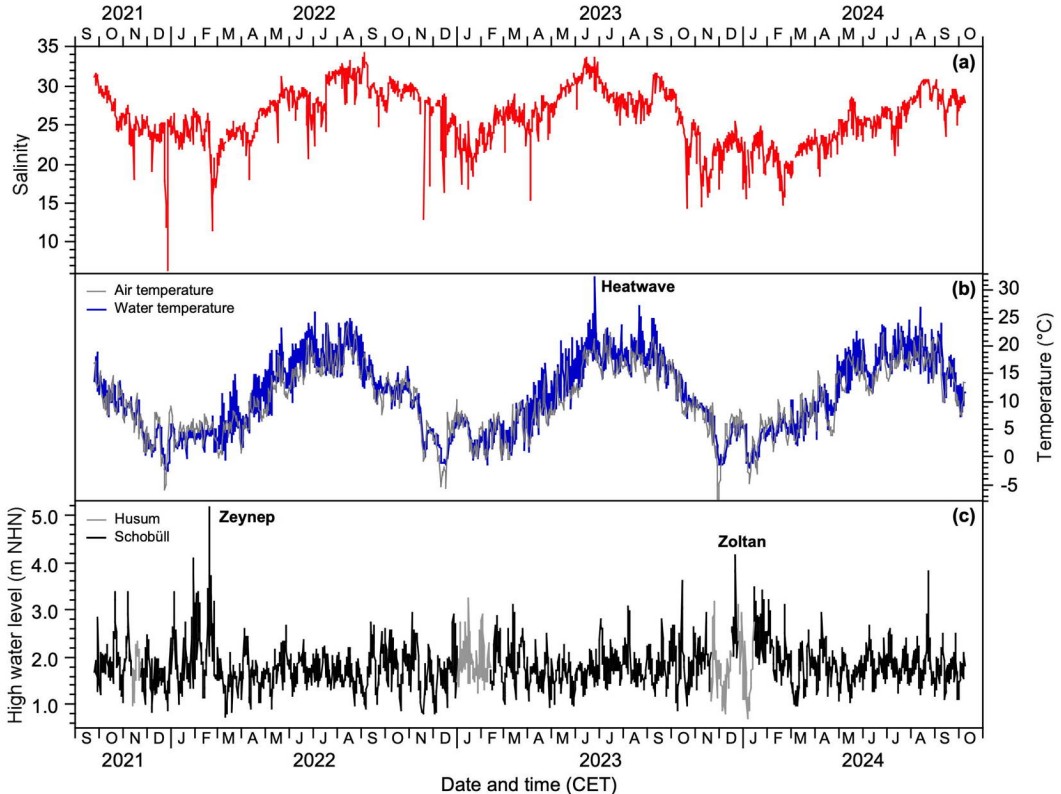

**Figure 3. High water salinity (a), temperature (b), and water level off Schobüll and at Husum (c). "Zeynep" and "Zoltan" refer to the two ranked storm surges during the investigation period.**

A regular pattern of spring and neap tides is clearly visible during spring and summer from March to September. Long term variations with high amplitudes during the equinoxes and low amplitudes in December and June were not recognised. The winter months from October to March were rather characterized by very strong, irregular fluctuations in high water levels (Fig. 3). Month-long phases of comparatively high water levels alternated with short intervals of very low levels. The mean high water level off Schobüll increased from 1.78 m NHN during the lunations 1222 to 1234, to 1.80 m NHN during the

lunations 1235 to 1247, and to 1.86 m NHN during the lunations 1247 to 1259. The mean high water levels exceeded the ten-year average of 1.72 m NHN at Husum tide gauge (1 November 2011 to 31 October 2020) in all three lunar years.

The salt marsh submergence times at the vegetation boundary at 1.03 m NHN increased from 33.1 % of the observation time during the 2021/2022 study period to 37.2 % during the 2023/2024 period. At the beginning of the lower salt marsh at 1.70 m NHN, the flooding time increased from 10.1 % in 2021/2022 to 14.0 % in 2023/2024. At the beginning of the upper salt

marsh at 2.20 m NHN, the submergence time increased from 2.2 % in 2022/2023 to 3.5 % in 2023/2024. The flooding time remained relatively constant at the top of the upper salt marsh at 2.50 m NHN, where it varied from 0.8 % to 1.4 % of the time in the three lunar years investigated.





During the entire study period, 19 storm surges were recorded. The two strongest events occurred during storm "Zeynep" on 19 February 2022, with a maximum water level of 5.18 m NHN, and during the storm "Zoltan" on 22 December 2023, with a maximum water level of 4.15 m NHN.

The salinities of the high waters off Schobüll showed a distinct seasonal pattern, with low and strongly fluctuating values in winter and steadily increasing but less variable values in summer (Fig. 3). After one or two maxima between July and September, the salinity decreased again during autumn. Smaller, multi day fluctuations are superimposed on the seasonal salinity cycle. These smaller fluctuations were found to match neap or spring tides in a few cases only. Particularly during the winter months, profound short-term minima were recorded, during which salinity drops to values as low as 6 units. Annual mean values ranged from $24.0 \pm 3.5$ units (1 sigma, n = 689) in 2023/2024 to $27.2 \pm 3.1$ units (1 sigma, n = 682) in 2022/2023. Maximum values varied between 30.8 and 34.2. The additional measuring point below the board walk of the old pier at 1.83 m NHN showed that the salinities of flood waters were on average $2.5 \pm 2.9$ units (1 sigma, n = 801) lower at the surface than at depth, as recorded at the regular measuring point at 0.94 m NHN.

The water temperatures showed profound seasonal cycles with minimum values of up to -2.6°C in December to maximum temperatures of 26.2 and 26.9°C in June to August. An exceptionally high temperature maximum of 32.5°C was recorded in June 2023 (Fig. 3). In autumn and winter, the water temperatures were in good agreement and often slightly lower than the daily mean air temperatures, although the latter showed a higher variability. Moderate to strong air frost only occurred during single, short periods in November to January. From March or April onwards, the high water temperatures were often higher than the daily mean air temperatures. The water temperatures showed marked differences between warm daytime and colder night-time floods, especially in spring and early summer (Fig. 3). From September onwards, these differences in the daily cycle diminish. The annual mean water temperatures were relatively constant at 10.8 to 11.4°C. The annual mean air temperatures showed an even lower variability at 9.9 to 10.2°C.

## 3.3 Extreme events

### 3.3.1 The Centennial Flood

A flood with an extreme high water level occurred in the western Baltic from 18 to 21 October 2023. The flood was the second-ranked storm surge since 200 years (Nöthel et al., 2024), and caused damages to the local infrastructure, dikes, boats, and homes in the order of 200 million € (Kieler Nachrichten, 2023). The flood was recorded by the P/T loggers at the measuring point in the terminal ditch of Bottsand lagoon up to a level of 1.69 m NHN, which corresponds to the height of the air valves of the assembly (Fig. 4). The water level rise and fall was almost synchronous with the record of Holtenau tide gauge, where the highest level was 1.95 m NHN on 20 October 2023, at 21:40 CET. The water temperature in the terminal ditch of Bottsand lagoon followed the usual diurnal pattern before and after the flood. During the flood event, the temperature followed the course of the salinity. The salinity showed two sudden increases (Fig. 4). The first increase took place on 18 October, after 15:00 CET, when the water level rose above 0.36 m NHN. The second salinity rise was recorded



shortly after the peak water level on 20 October, at 22:40 CET. The first rise can be linked to the submergence of the
extensive salt marshes at that level. The second rise can be related to the flooding of the beach ridges and former point bars
in the nature reserve, which are about two m high.

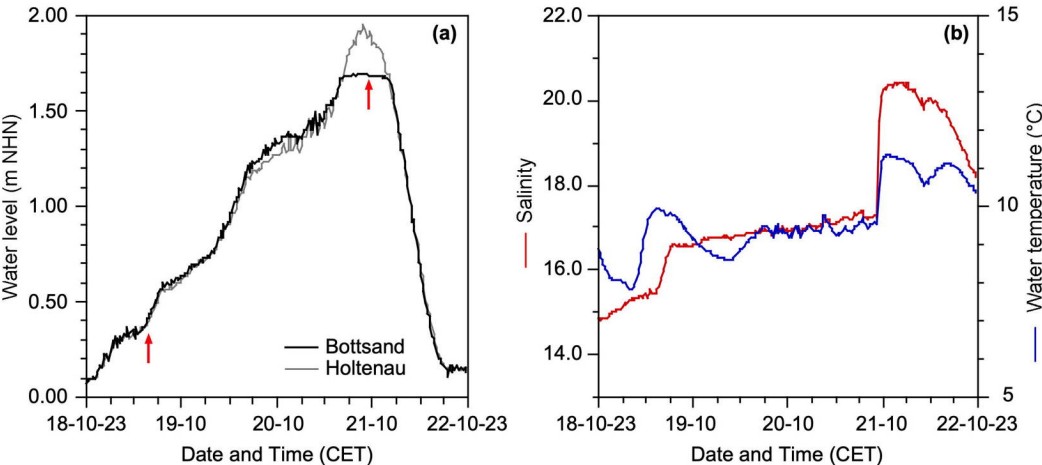

**Figure 4. Water level (a), temperature, and salinity (b) during the Centennial Flood at Bottsand and Holtenau. The red arrows in
(a) mark the salinity rises.**

### 3.3.2 Storm surges Zeynep and Zoltan

The storm surge "Zeynep" occurred on 19 February 2022. A peak level of 5.10 m NHN was recorded by Husum tide gauge
and 5.18 m NHN off Schobüll, i.e. 3.46 m above mean high water level at Husum. "Zeynep" was as high as the Great Hallig
Flood on 3 to 5 February 1825 and thus is recognised as a significant event. A comparison of the water level data from
Schobüll and the tide gauge record from Husum revealed that the tide raised earlier and fell almost simultaneously off
Schobüll (Fig. 5). During the high waters after "Zeynep", the tide raised earlier and fell later off Schobüll than at Husum.
The constant water level measurements off Schobüll during low tide were due to a large tidal pool that was situated before
the western end of the groyne of our measuring point. The waters could not flow out completely due to the wind stress, and
the level in the tidal pool was even higher by about 0.1 m after the storm surge (Fig. 5). The salinity of the storm surge water
was in line with a longer trend of successively decreasing values. In detail, the salinities increased during each flood until a
maximum around high water and decreased again when the water level fell again. The high water with a markedly low level
before the storm surge followed an exceptionally low ebb period. This high water showed an irregular salinity pattern and a
peak value lower by three units than the main surge (Fig. 5). The water temperatures showed a similar pattern as the salinity
values, in particular at night time when the air temperatures were lower than the water temperatures.

The storm surge "Zoltan" showed a double high water maximum on 21 and 22 December 2023. They reached a water level
of 3.83 and 4.15 m NHN off Schobüll, which was 0.18 and 0.22 m higher than at Husum tide gauge and 2.11 and 2.43 m



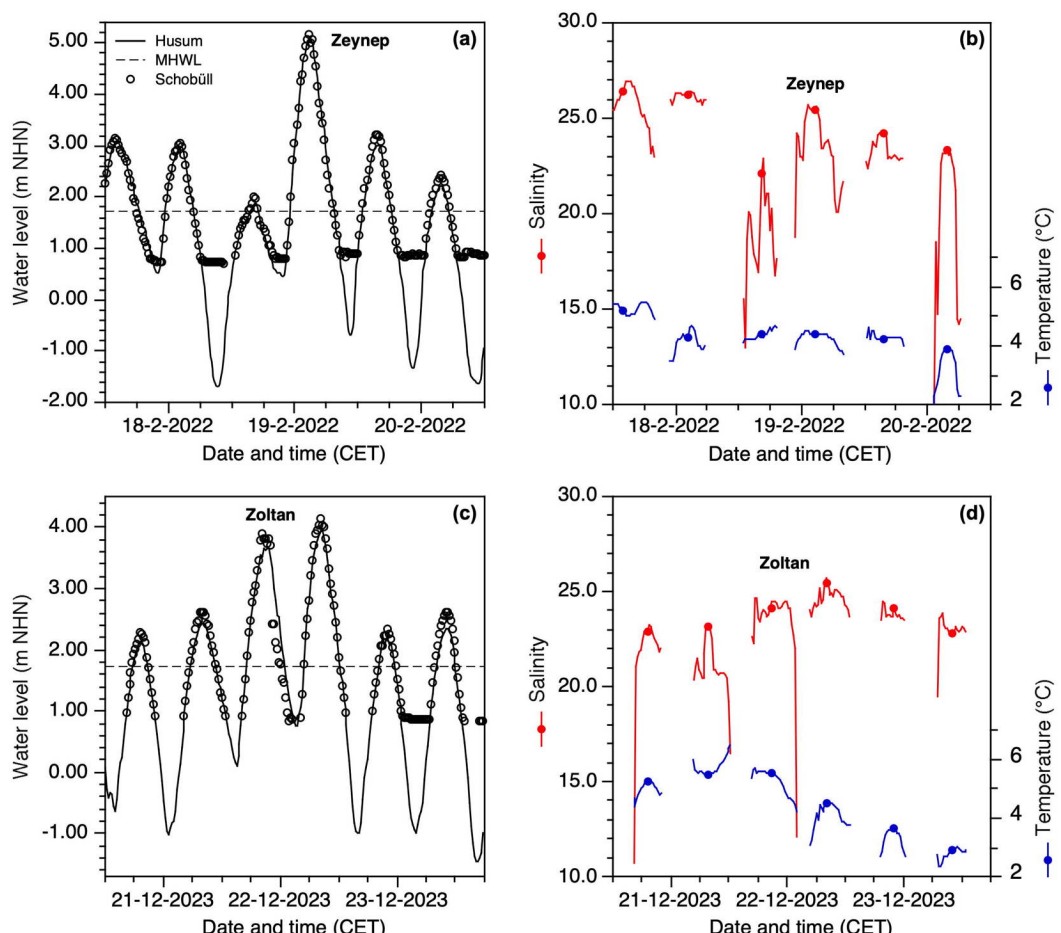

**Figure 5. Water level (a, c), temperature and salinity (b, d) during the storm surges "Zeynep" and "Zoltan" off Schobüll and at Husum (water level "Zeynep" after Bundesamt für Seeschiffahrt und Hydrographie, 2023). The dots mark the measurements of**
395 **temperature and salinity at high water times.**

above mean high water at Husum. "Zoltan" was a less severe event in the historical context of storm surges on the North Sea coast. While the water level raised almost simultaneously and fell slightly later during the second, higher surge off Schobüll as compared to Husum, it fell much earlier during the first surge after a sudden drop in water level off Schobüll (Fig. 5). During the preceding and following high waters, the water level raised earlier and fell later off Schobüll with reference to

400 Husum tide gauge records, similar as after "Zeynep" storm surge. The tidal pool off Schobüll impeded the backflow of ebb waters on the 23 December 2023 only, after the main storm surge. The salinity of tidal waters followed a longer trend with a broad maximum during the second surge of "Zoltan" (Fig. 5). The individual dynamics of rising salinity values until a maximum shortly after high water and a decrease afterwards was less developed than during the high waters around "Zeynep". The high water before the first surge of "Zoltan" showed a stronger, internal variability and reached a lower peak

value than during the next high water. The temperature curve revealed that the surficial flood water was warmer than at





depth during this particular event, which was preceded of an exceptionally low ebb tide (Fig. 5). During all other high waters around "Zoltan", the surficial waters were cooler than at depth.

### 3.3.3 North Sea heat waves

Three events were recognised in summer 2023 when the temperature of tidal water reached or exceeded 25°C, namely from 22 to 25 June, 20 to 22 August, and 6 to 8 September (Fig. 3). These high waters occurred in late afternoon between 16:00 and 19:20 CET. The preceding high waters, which occurred early in the morning, were cooler by 7.8 K on average. The mean temperature difference between day and night high waters, or those in the morning and in the evening, was 2.6 K before and after the warm events. The water temperatures followed the trend of the air temperature in the salt marsh (Fig. 6).

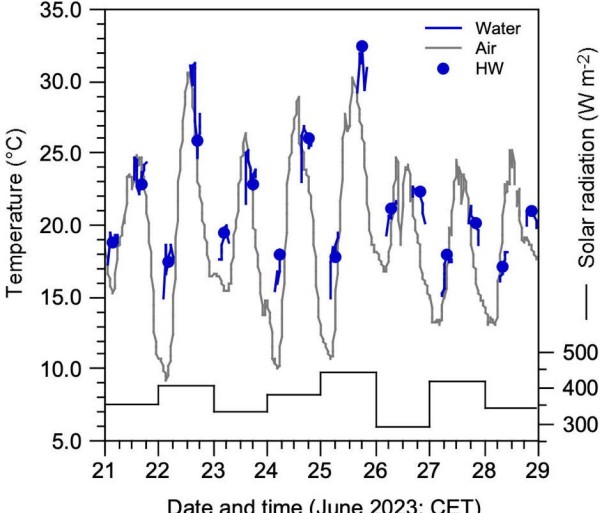

**Figure 6. Temperatures and solar radiation during the North Sea heat wave off Schobüll in June 2023 (solar radiation after https://www.schobuell-wetter.de). The dots mark the water temperature measurements at high water times.**

They were same as high in the morning and much warmer in the afternoon or at night. In particular during the first event in late June, the water temperatures exceeded the maximum air temperatures on days with a high solar radiation, i.e., with a long period of sunshine (Fig. 6). No covariance with the high water level, spring or neap tide intervals was recognised.

## 4 Discussion

### 4.1 Seasonal dynamics of temperature and salinity in Bottsand lagoon

The lagoonal temperatures showed a clear seasonal cycle. They followed the air temperatures in winter, and were higher than the air temperatures in spring and summer. The lagoonal temperatures are compared with the surface water record from the





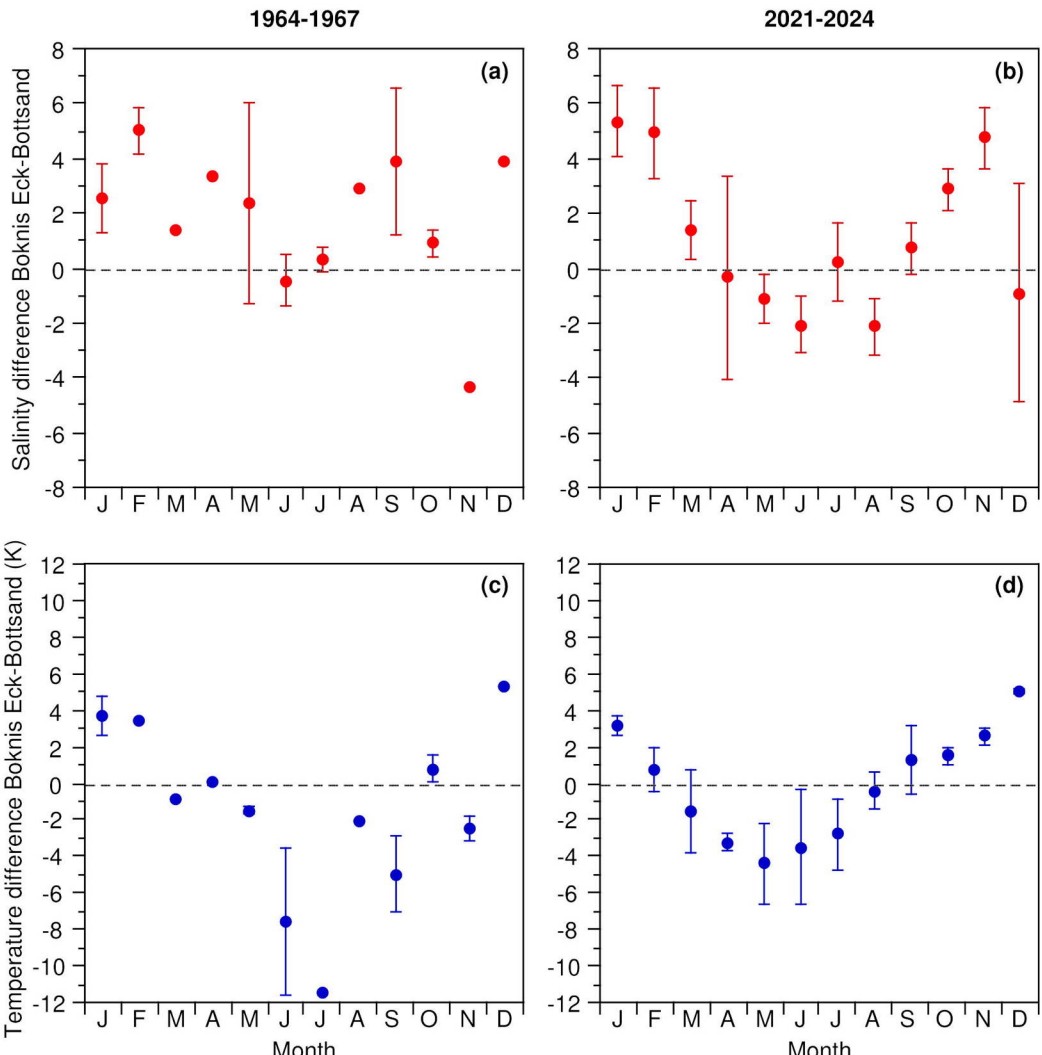

**Figure 7. Mean monthly salinity (a, b) and temperature (c, d) differences between Bottsand lagoon and Boknis Eck. Dots: mean values, error bars: range of values.**

Boknis Eck time series station northwest of Bottsand, which served as a reference for the conditions in the southwestern Baltic Sea (Bange et al., 2011). The seasonal temperature cycle at Bottsand lagoon during the 2021 to 2024 investigation period was in agreement with the seasonal variability observed at Boknis Eck (Bange et al., 2011; Lennartz et al., 2014), even though the Boknis Eck time series has a monthly resolution (Supplement Table S7). On the Boknis Eck sampling day, the daily mean temperatures from Bottsand lagoon were higher by 0.4 to 4.4 K than in the surface water at Boknis Eck from March to August. In autumn and winter, the surface temperatures were up to 5.1 K higher at Boknis Eck than at Bottsand lagoon (Fig. 7).

The salinities at Bottsand showed no seasonal cycle but one- or two-months periods of higher values in November, December and March, and low values in January, February, April and May. Bottsand is the westernmost in a series of



lagoons on the southern Baltic coast. Their hydrology is considered to be governed by freshwater influx, precipitation and evaporation, and seawater incursions (Lehmann et al., 2022). In general, the salinity is lower and the water level is higher in the lagoons than in the adjacent Baltic Sea. This is also mirrored by the mean water level of Bottsand lagoon, which was 0.11 m higher during the investigation period than the 10 years mean water level at Holtenau tide gauge. The salinity of
Baltic Sea surface water is mainly driven by freshwater discharge of the major rivers shedding into the central and eastern Baltic Sea (Bergström and Carlsson, 1994; Rohde and Winsor, 2002), which usually peaks in early spring. Consequently, the surface water at Boknis Eck shows a pronounced salinity minimum in April to June (Lennartz et al., 2014, there Fig. 3). At Bottsand lagoon, a short minimum in April and May was recognised. None-the-less, the salinities were higher by 0.3 to 2.1 units at Bottsand from April to August during the 2021 to 2024 investigation period. In winter, the salinity was higher by up
to 5.4 units at Boknis Eck (Fig. 7). This seems to be contradictionary as Bottsand lagoon was flushed more often by high waters during the cold season. The salinity record from the Centennial Flood depicted in detail how the high water replenished the salinity of the lagoon by the inundation of barriers, and that the effect of this flush was of limited duration. The salinity maximum was counterbalanced by rainfall or groundwater seepage within days. Earlier investigations revealed that short term, tidally driven water level fluctuations and evaporation at high temperatures were crucial for the salinity of
Bottsand lagoon, while the diluting effect of ground water seepage was also recognised (Lutze, 1968; Schönfeld, 2018). Situation was different in the mid 1960s (Lutze, 1968, there Fig. 19). The mean temperatures at Bottsand lagoon were lower by 0.4 K and salinities were lower by 0.6 units during two annual cycles from December 1964 to January 1967 than during the three annual cycles from early October 2021 to late September 2024 that were investigated in the present study (Fig. 7). At Boknis Eck, the surface water temperatures were on average 1.2 K lower in the mid 1960s than in the early 2020s, while
the salinity was lower by 0.1 unit. These differences are in agreement with earlier estimates of 0.2 K warming per decade, while the Baltic Sea surface water salinity did not change during the 20[th] century (Lennartz et al., 2014). In the mid 1960s, the temperatures in Bottsand lagoon were consistently higher than at Boknis Eck by 1.5 to 11.5 K from May to September, hence the warm season was delayed by one month as compared to the early 2020s. The salinities were mostly higher at Boknis Eck than at Bottsand lagoon in the mid 1960s, with an outlier in November. Only in June and July, the salinities at
Boknis Eck and Bottsand lagoon were similar. The differences of Boknis Eck and Bottsand temperature and salinity showed much more scatter in the 1960s than in the 2020s, which allowed only limited conclusions. The scatter may be due to the fact that the temperatures from Bottsand were single, hand held measurements or water samples, and that they were not taken on the same day as at Boknis Eck (Supplement Table S8). The average time difference of sampling at both locations was six days. On the other hand, the geographical setting and lagoonal bottom morphology was different before Marina Wendtorf
and the adjacent resort was built in 1972 (Gemeinde Wendtorf, 1990). The access to the lagoon from the Baltic Sea was much wider before the construction, and a shoal separating the inner lagoon from the marina and port entrance was not present in 1966 (Lutze, 1968, there Fig. 1; Schönfeld, 2018). In particular the shoal is deemed pivotal for the salt enrichment in the inner lagoon during the warm season, which is effective today.




## 4.2 Temperature and salinity dynamics in Husum Bight

The water temperatures recorded in Husum Bight off Schobüll showed a seasonal cycle and followed the air temperature record. The water temperatures were often lower than the air temperatures in winter and higher in spring and summer. The high waters were warmer during the day than at night-time in spring and early summer.

The high water temperatures off Schobüll during the 2021 to 2024 investigation period were supplemented by earlier measurements since 2014. The record displayed the same pattern of seasonal cycles as in Lister Ley off Sylt (Rick et al.,

2023). Individual, short term fluctuations showed a higher amplitude off Schobüll than off Sylt (Fig. 8). They may be attributed to the differences between day and night time high water temperatures, at least in spring and summer. The 2023 heat waves were not recorded in Lister Ley, neither a similar event of 25.4°C on 8th September 2016. This may be due to the

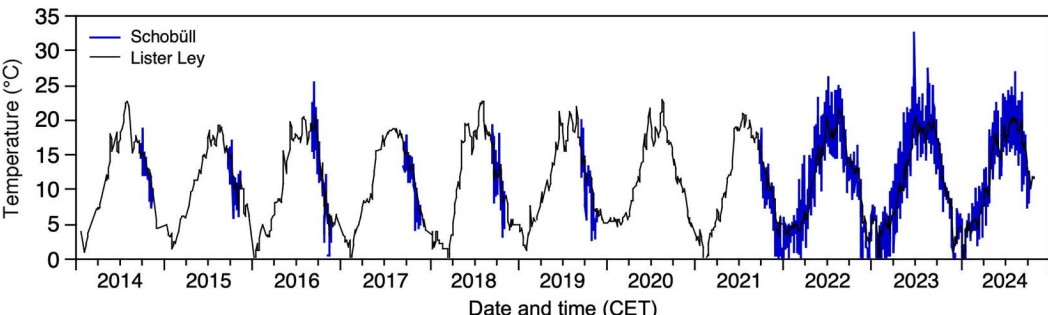

**Figure 8. High water temperatures off Schobüll and surface water temperatures at Lister Ley monitoring station during the years 2014 through 2024.**

fact that the exceptionally high flood temperatures were recoded off Schobüll in late afternoon while the measurements in Lister Ley surface waters were taken in the morning. The air temperatures above the salt marsh off Schobüll showed no response to the submergence with flood waters. Instead, the water temperature rather followed the trend of the air

temperature, in particular in the morning and in the afternoon (Fig. 6). It is therefore conceivable that the heat waves were mainly effected by air temperature and amplified by solar radiation. Conductive heating by the warm surface of the dark-coloured tidal flats could also have contributed to the flood water temperatures in the afternoon. However, sediment temperatures were not taken to further constrain their influence.

An increase in temperatures over the years, which has been indicated as a trend toward higher winter temperatures in Lister

Ley since 2017 (de Amorim et al., 2023), was not recognised off Schobüll. Instead, we recorded an increase in



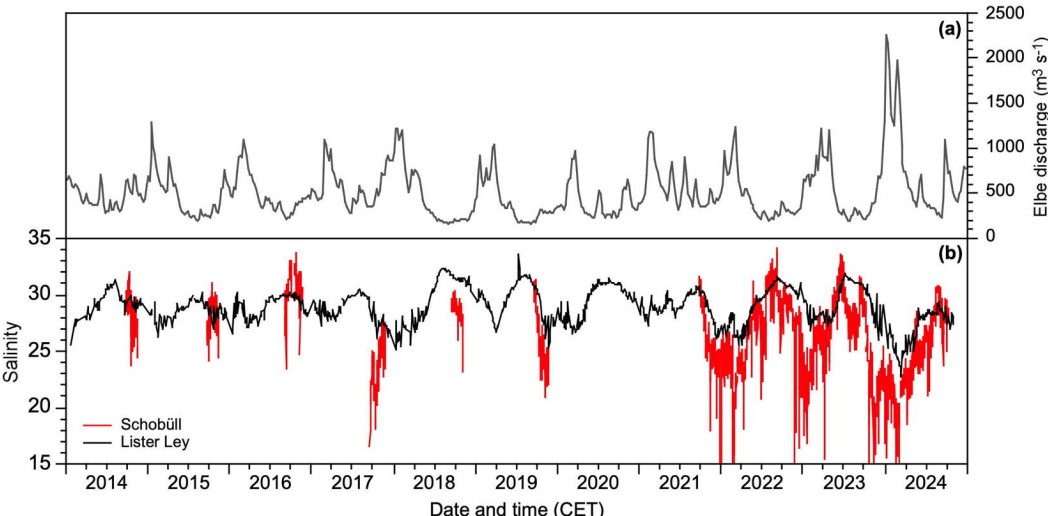

**Figure 9. Weekly average Elbe river discharge at Neu Darchau gauge (a; by courtesy of Wasserstraßen- und Schifffahrtsamt Elbe, Magdeburg, Germany), high water salinities off Schobüll, and surface water salinities recorded at Lister Ley monitoring station (b) during the years 2014 through 2024.**

annual mean temperatures by 0.6 K from the 2021/2022 to the 2022/2023 investigation period. This increase is in good agreement with the 0.4°C rise in the mean surface temperature of the North Atlantic from 2022 to 2023 (Kuhlbrodt et al., 2024, there Figure 1). The covariance could be accidental, because an increase in the mean surface water temperature from the same 2021/2022 to the 2022/2023 investigation period was not recognised in Lister Ley. The mean water temperatures decreased by 0.1 and 0.2 K in the 2023/2024 period off Schobüll and in Lister Ley. The mean air temperatures at the private weather station Schobüll were 9.9°C, 10.1°C, and 10.2°C during the same periods, a variability, which neither fits to the water temperatures off Schobüll nor in Lister Ley.

The salinities off Schobüll exhibited a profound seasonality during the 2021 to 2024 investigation period. High salinities prevailed in summer and low values in winter. The same seasonality was observed in Lister Ley, where the salinities were on average 2.7 units higher than off Schobüll (Fig. 9). They were also higher by 0.6 to 3.9 units in Lister Ley than off Schobüll in autumn 2014, 2017, and 2018. The seasonlity of salinity variations in Lister Ley from 2014 to 2024 is mirrored by the Elbe river runoff, which is deemed to control the salinity in the south-eastern German Bight (Fig. 9). Discharge maxima occurred during the months December to April (Klein and Frohse, 2008), while the strongest peak in the last 140 years happened in June 2013 (Merz et al., 2014; Voynova et al., 2017). The fluvial runoff effected that the waters were haline stratified throughout the year off the Weser and Elbe estuaries, whereas the waters were vertically mixed in the Wadden Sea throughout the year (Frey and Becker, 1987). Even there, a pervasive gradient of landward decreasing salinities was recognised (Janssen et al., 1999), which was mirrored in the offset between Husum Bight and Sylt Roads tidal waters. A cross correlation between weekly average salinities in Lister Ley and off Schobüll for the 2021 to 2024 investigation period showed the highest correlation coefficient of r = 0.843 (p = 4.5 x 10$^{-43}$) at a lag of three, i.e. it takes more than 20 days until a



water body of lower salinity is transported from the Heverstrom tidal channel system northwards to Lister Ley. Similarly, a
cross correlation between the weekly averaged Neu Darchau gauge Elbe discharge record and the salinities off Schobüll
showed the most significant correlation of r = -0.592 (p = 1.3 x $10^{-15}$) at a lag of seven, which suggested that it takes more
than 45 days for an Elbe river freshwater pulse to reach Husum Bight and Schobüll. This time lag compares well to the
spreading of the Elbe peak discharge in June 2013, which was recognisable after 14 days off Büsum, and covered the entire

south-eastern German Bight up to the isle of Helgoland 28 days after the discharge event (Voynova et al., 2017).
Short-term variations and the amplitude of seasonal fluctuations were higher off Schobüll than in Lister Ley (Fig. 9). While
salinity levels remained constant there over the last 10 years, they have decreased by an average of 2.9 units off Schobüll.
This decrease in salinity is unlikely to be due to increasing freshwater input from the rivers, because their discharge actually
decreased during the same period (Philippart et al., 2024). It neither mirrors an annually increasing precipitation, which

showed no consistent trend at the private weather Station Schobüll during the last decade. It is yet unclear to which extend
groundwater seepage results in regionally varying salinities. At the edge of the moraine hill Schobüll Geest, the influence of
groundwater seepage on the flora of the salt marsh has been documented (Stock, 2013). This water drained along the ditches
at low tide and spread across the mudflats in front of the vegetation boundary at low tide, where a salinity of 6 to 9 has been
measured by the author on unnumbered visits at Schobüll. Another evidence for a local, low-salinity lens on top of tidal

waters was provided by continuously rising and falling salinity levels at the onset and termination of high tides. Furthermore,
exceptionally low salinities were recorded after profound low tides before storm surges (Fig. 5), during which the stronger
hydraulic gradient between drainage ditches and sluices and the receiving waters in tidal channels effected a higher
freshwater runoff.
Evidence for another influencing factor was provided by a comparison of the daily mean salinities with the daily

precipitation measured at the private weather station in Schobüll. Most short-term salinity minima coincided with periods of
high precipitation lasting several days (Fig. 10). A cross-correlation of salinities and precipitation yielded the highest
correlation coefficient of r = -0.185 and the highest significance level (p = 6.8 x $10^{-10}$) at an offset of six. This means, the
salinity decreased substantially about one week after persistent high precipitation or heavy rain events at Schobüll. This time
lag is in good agreement with data from the Netherland's Wadden Sea, where a time lag of 5.5 days has been observed

between discharge flow rate from the outlet sluices of Kornwerd and Den Oever and the salinity in the Marsdiep
(Zimmermann, 1976; van Aken, 2008). As in the Netherland's Wadden Sea, the high water salinity off Schobüll increased
again a few days after the precipitation stopped, presumably due to mixing processes in the tidal current.



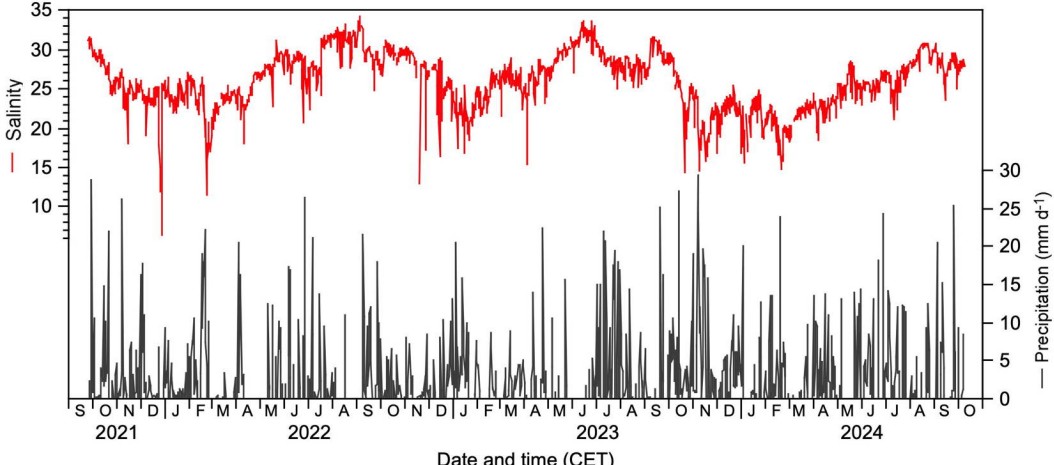

**Figure 10. High water salinity off Schobüll and daily precipitation recorded at the private weather station Schobüll (after**
**https://www.schobuell-wetter.de).**

### 4.3 Boundary levels of salt marsh vegetation zones

The ecology of salt marshes was often related to flooding frequencies, i.e., number of submergence events per observation time (e.g. Nolte et al., 2013; Lange et al., 2019; Reents et al., 2021). Inundation frequencies, i.e. cumulative submergence time per period of observation, were rarely reported in the literature, even though they provided a more comprehensive

picture (Silvestri et al., 2005; Li et al., 2018; Schönfeld and Mendes, 2022). The inundation frequencies were very variable at both, Bottsand and Schobüll during the 2021 to 2024 investigation period (Supplement Table S6). They ranged from 60.6 to 76.1 % of the one lunar year observation time at the base of the lower salt marsh vegetation zone with *Puccinellia maritima* at Bottsand, and 10.1 to 14.0 % at Schobüll. The base of the upper salt marsh vegetation zone with *Festuca rubra* was submerged during 8.5 to 17.9 % of the time at Bottsand and during 2.7 % to 3.5 % of the time at Schobüll. Even though the

same halophyte species are concerned, the cumulative submergence times were longer at Bottsand than at Schobüll. The general difference of water level variability between both study areas was a microtidal regime and an irregularly submerged salt marsh at Bottsand, while the regime was mesotidal and the salt marsh was regularly flooded twice a day at Schobüll. As the average salinity was higher at Schobüll (25.9 units) than at Bottsand (15.9 units), the differences of inundation frequencies suggest that a certain salinity of pore waters has to be maintained in the soils, on a long term and against the

counteracting dilution by precipitation (Costa et al., 2003), to sustain the lower and upper salt marsh floral associations. It has to be emphasised that halophytes are adapted to but do not require a certain soil salinity for growth or reproduction. Under non-saline conditions, the salt marsh species are not competitive. Their adaptive mechanisms give them an advantage to sustain against other plants at higher soil salinities. On the other hand, high salinities and long submergence times may impede the seed germination of halophytes (Phleger, 1971; Baldwin et al., 1996; Elsey-Quirk et al., 2009).

The lower boundary of the pioneer vegetation zone was submerged during 33.1 to 37.2 % of the time at Schobüll. These figures are in agreement with the outward limit of *Spartina alterniflora* along the Jiangsu coast in China, which was flooded



during 32 to 40 % of the observation time (Li et al., 2018). Higher inundation frequencies of the vegetation boundary were reported from North Inlet, South Carolina, USA (43 %) or from the Virginia Coast Reserve, USA (51 %; Li et al., 2018). The base of the pioneer vegetation zone was found to occur successively lower in the tidal frame at higher tidal amplitudes

(Balke et al., 2016), which suggests a longer submergence period during each tidal cycle. However, the tidal ranges are 1.4 and 0.8 m at North Inlet and in the Virginia Coast Reserve, and thus much lower than in China (3.0 to 4.5 m) or at Husum tide gauge close to Schobüll (3.51 m). These figures corroborate the results from Patos Lagoon, Brazil (Costa et al., 2003), that a longer submergence time is necessary at low water level variations to sustain the salt marsh pioneer vegetation. In a global scope and despite the settings in Patos Lagoon, submergences times in the pioneer vegetation zone were seldomly

higher than 50 %. It has been invoked that the vegetation boundary rather matches the Mean Neap Tide level (Adam, 2002). This level offers a reasonable chance of a couple of days without submergence during which seedlings of *Spartina* spp. can dry, germinate, and anchor in the substrate (Balke et al., 2014). Off Schobüll, the Mean Neap Tidal level varied from 1.53 to 1.66 m NHN, which is much higher than vegetation boundary at 1.03 m NHN. These data have to be taken with caution, however, because not all neap tides were recorded during the 2021 to 2024 investigation period of the present study. Another

proliferation mechanism of *Spartina* spp. is clonal integration, which even enhances the flood tolerance of daughter ramets (Xiao et al., 2010).

### 4.4 A biotic response to extreme events

*Crassostra gigas* (*Magallana gigas* of authors) was cultured off List, Sylt, since 1986, spread in 1991, and was recognised as invasive species in the Wadden Sea thereafter (Nehls et al., 2006). The first specimens were recorded off Schobüll in

November 2011 (own observations). Empty shells and single, living specimens were occasionally found under the old pier and before the seaward groynes since 2019. A mass occurrence of minute *Crassostrea gigas* shells was observed in spring 2024 in the sandy mud before the vegetation boundary off Schobüll. The low number of growth stripes and a shell size of less than half a cm inferred that these oysters have grown for some months only. A temperature of 23 to 25°C has been considered as optimal for larval development and recruitment of this species (Quayle, 1988; Kobayashi et al., 1997). As

such, it is conceivable that a successful proliferation event has been taken place during the heat waves in summer 2024, similarly as in the early 2000s off Sylt (Diederich et al., 2005). The reason why they all died off is probably rooted in their salinity requirements: oyster embryos can not tolerate salinities of less than 18 units (Wiltshire, 2007). Such low salinities were recorded off Schobüll in November 2023 and, more importantly, between 16 and 23 February 2024, after the peak Elbe river discharge event in early January 2024 (Fig. 3).

### 5 Conclusions

This paper presents the first continuous time series of water level, temperature, and salinity from Bottsand lagoon at the Baltic Sea coast, and from the mudflats off Schobüll at the North Sea coast of Schleswig-Holstein, covering three annual



cycles from September 2021 to October 2024. The data characterised the properties and inundation frequencies of tidal waters submerging the salt marshes at high temporal resolution. A short term variability and extreme events were captured, which were not recorded by the off-shore time series stations Boknis Eck in the southwestern Baltic Sea and Sylt Roads in the North Sea. The initial hypothesis is therefore rejected.

The deployment of hydrographic instruments in intertidal environments in an appropriate, environmentally friendly manner imposed technological challenges and compromises, which were also reflected in data coverage and measurement accuracy. The mean difference of on-site manually measured and data logger recorded values was 0.03 m for water depth, 0.3 K for temperature, and 0.5 salinity units, an accuracy which is deemed acceptable with reference to the profound environmental dynamics observed.

At Bottsand lagoon, the temperatures followed the air temperatures in winter, and were higher than the air temperatures in spring and summer. The salinities did not follow a seasonal cycle. Instead, they showed one or two months periods of consistently higher or lower than average values in winter and spring. The lagoon showed an individual dynamics in comparison to Boknis Eck, where the temperatures and salinities were lower in summer and higher in winter. The seasonal salinity differences were less developed in the mid 1960s, when the connectivity of the lagoon with the Baltic Sea was less constricted and a sandy shoal separating the inner part of the lagoon from the Marina Wendtorf port entrance was not present. The key function of this shoal for the salt enrichment in the inner lagoon during the warm season was corroborated by the present study. In particular, the salinity record of the Centennial Flood in October 2023 depicted discrete salinity rises when certain barriers were flooded.

In Husum Bight off Schobüll, water temperatures were often lower than the air temperatures in winter and higher in spring and summer. The high waters were warmer during the day than at night-time in spring and early summer only. Observations made during the major North Sea heat wave in June 2023 revealed in detail how the water temperature followed the air temperature and how it was further amplified by solar radiation. Supplementary, conductive heating by the warm, dark-coloured tidal flats must not be excluded. The salinities off Schobüll were higher in summer and lower in winter. The seasonal cyclicity was related to the Elbe river runoff, which largely influences the salinity in the south-eastern German Bight. Discharge maxima occurred during the months December to April. The same seasonal cycle was recorded in the Sylt Roads time series. Cross-correlations of the records revealed that it takes seven weeks for an Elbe river freshwater pulse to reach Schobüll in Husum Bight, and three weeks more to proceed to the Lister Ley tidal channel system off Sylt. The salinities were on average 2.7 units lower off Schobüll than off Sylt. This offset mirrors a pervasive gradient of landward decreasing salinities in the Wadden Sea. The detailed records obtained during the storm surges "Zeynep" and "Zoltan" in February 2022 and December 2023 and other observations provided compelling evidences for the presence of a local, low-salinity lens on top of tidal waters, which is not sufficiently constrained to date. It may be fed by groundwater seepage or by freshwater runoff through drainage ditches and sluices in a wider area. A cross correlation with the precipitation record from Schobüll weather station revealed that the salinity decreased substantially about one week after persistent high precipitation or heavy rain events. This time lag is in good agreement with data from the Netherland's Wadden Sea.



The salt marsh inundation frequencies, i.e. cumulative submergence time per period of observation, were very variable in both locations among the three years. At the lower boundaries of the lower and upper salt marsh vegetation zones, the inundation frequencies were consistently higher at Bottsand than at Schobüll, even though the same halophyte assemblages

were present in both areas. As the average salinity was 10 units higher at Schobüll than at Bottsand, the differences of inundation frequencies suggest that a certain salinity has to be maintained in the soils to sustain the lower and upper salt marsh floral associations. The high inter-annual variability of submergence times seems to be tolerated by the perennial plants. A longer monitoring and detailed investigations of sedimentary pore waters are necessary to further constrain this conclusion.

A mass occurrence of small Pacific oyster shells before the vegetation boundary off Schobüll was observed in spring 2024. With reference to literature data, it could be related to an oyster spatfall triggered by water temperatures of more than 23°C during the North Sea heat waves observed off Schobüll in summer 2023. The reason why they all died off after half a year is probably related to a short period of salinities lower than 18 units in late February 2024, after the Elbe river peak discharge event in early January. These unfortunate circumstances in course of environmental extremes demonstrated how vulnerable

the Wadden Sea biota are, and that it is a matter of resilience of both, fauna and flora, to sustain at times of Global Change.

**Data availability**

All raw data are made available through the open-access data repository PANGAEA. This also includes data retrieved from online resources. Derivated data are provided in the Supplement.

**Supplement**

The supplement related to this article will be made available online.

**Author contributions**

JS conceptualised the study, performed the data collection and conducted the fieldwork. HWB, HH and SR contributed further data sets to the study. JS conducted the data analysis and wrote the original draft. All authors were involved in the interpretation of the results and the writing of the manuscript.

**Competing interests**

Hermann W. Bange is a member of the editorial board of Biogeosciences.



## Acknowledgements

Christian Wiedemann (Landesbetrieb für Küstenschutz, Nationalpark und Meeresschutz Schleswig-Holstein, Nationalparkverwaltung Tönning), Marieke Ruge (Amt für Umwelt, untere Naturschutzbehörde des Kreises Plön), and Carsten Harrje (NABU Schleswig-Holstein, Fachreferent für das NSG Bottsand) permitted access to the conservation areas. Anke Dettner-Schönfeld helped during fieldwork and did the levelling. Daniela Supper-Nilges (Bundesanstalt für Gewässerkunde, Koblenz) and Mark Raschewski (Wasserstraßen- und Schifffahrtsamt Elbe, Magdeburg) provided tide gauge data from Husum and Elbe river discharge data from Neu Darchau gauge, which were not accessible online. Sonja van Leeuwen (NIOZ, Texel, The Netherlands) gave advice on riverine influx in the North Sea, which is gratefully acknowledged. The Boknis Eck Time-Series Station is run by the Chemical Oceanography Research Unit of GEOMAR. Helmke Hepach was supported by the BMBF-funded project "CREATE" (project number: 03F0910A; research mission "sustainMare" of the Deutsche Allianz Meeresforschung). We thank Hans-Peter Hansen, Frank Malien and Kastriot Qelaj for providing CTD data from Boknis Eck. We thank the captains and the crews of the FK Littorina and FB Polarfuchs.

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
