# Peer review of "Hydrography of intertidal environments in Schleswig-Holstein, Germany"

_EGUsphere, 2025_

## Author Response (AR1)

**MS No.: egusphere-2025-2672, posted as preprint on EGUsphere for public review and discussion on 7 Aug 2025**

**Title: Hydrography of intertidal environments in Schleswig-Holstein, Germany.**

**Authors: Joachim Schönfeld (jschoenfeld@geomar.de), Hermann W. Bange, Helmke Hepach, and Svenja Reents**

**Response to Reviewer comments**

Dear Editor,

we sincerely thank the Associate Editor Perran Cook, two anonymous Reviewers and Julia Lübbers for their constructive comments and suggestions. We acknowledge that they recognise the subject and relevance of our paper. The referees noted several points needing clarification and emendation.

We have addressed all the issues, include a point-by-point response to all the comments and questions, and provide a revised manuscript that we believe is much improved. In the following, the Reviewers' comments or questions on the manuscript are given in black, and our response is highlighted in blue and indented.

**Associate editor decision, Perran Cook**

The authors have responded constructively to 2 reviewer, and 1 general comment. I ask the authors to revise the manuscript as outlined in response to the reviewer comments. Of key importance is to improves the clarity and conciseness of the manuscript.

> Reply: we thank the Associate Editor for his encouragement. We have addressed all issues with particular emphasis on briefness and clarity of the Abstract, Methods, and Conclusions chapters.

**RC1: 'Comment on egusphere-2025-2672', Anonymous Referee #1, 17 Sep 2025**

'Hydrography of intertidal environments in Schleswig-Holstein, Germany'- submitted by Schönfeld et al., provided important temporal dataset of basic parameters of water and air in this study. Using the new dataset, along with a compilation of published dataset of 1960s, this manuscript showed short-term and long-term changes in the marginal marine environment.

> Reply: we are grateful for the thorough review and in particular appreciate the recognition of relevance of our contribution.

Although the manuscript provided incredible amount of dataset, in my opinion, the manuscript has not been presented well. Abstract and conclusion contain too many texts; it needs to be concise. Also, discussion and conclusion do not provide significant outcome instead of having extensive dataset.

> Reply: indeed, we created a large data set. As anything was new and many unexpected features were observed, they need to be documented and reported in text and figures. This is a matter of case for any baseline study. We concede that both, Abstract and Conclusions chapters can be shortened without loss of evidence and information. The Abstract has been formulated more concise in the revised version. It now comprises 2845 instead of 3477 characters.. The Conclusion chapter has also been stripped from non-essential information. It now comprises 4663 instead of 5704 characters.

The materials and methods section is very important as throughout the manuscript data used were obtained from sensors and corers. So, it's wise to explain the accuracy and precision of each parameter and how it was determined. However, this section is little bit messy and inconsistent in many aspects. Please see my comments below. I suggest summarizing all analytical data (accuracy, precision etc) for all parameters in a table.

> Reply: we concede that the middle part of Chapter "2.2 Hydrographical measurements" between lines 161 and 192 of the submitted version was difficult to follow. We therefore re-structured this part in that we first reported the measuring points, then described the assembly, and after that the operation procedures in the vised version of the manuscript. The sensor accuracy is reported in a following paragraph. A new Table is not deemed necessary as only three parameters were measured. Corers were not applied.

Para 160, first sentence: What is the relevance of this sentence?

> Reply: it is deemed necessary to report the devices we have used.

Explain what P/T and C/T loggers are. What does it mean by P, T and C.

> Reply: The sentence was reformulated in order to better explain the abbreviations: "As in earlier studies, Odyssey® C/T and P/T data loggers (Dataflow Systems Ltd., Christchurch, New Zealand) were used to measure water level (P), temperature (T), and salinity (C)."

Para 175, 180, first sentence: Cite Figure 1

> Reply: we respectfully disagree. The map in Figure 1 is too low in resolution to display the setting of the measuring points.

Para 200. How were accuracy determined? It's not clear to me how accuracy is shown by range and average. Please explain? What's the unit of salinity? I assume it is a practical salinity unit. Please clarify. Accuracy is generally reported in percentage.

> Reply: the sensor accuracy was determined under laboratory conditions, which is explained in the sentences before. The salinity is dimensionless and has to be expressed as "units". This has been already discussed in an earlier paper (Schönfeld, 2018, there page 385). We have added this reference here. Reporting the accuracy as percentage is not correct in our case, because no single-point calibration was applied.

Para 205: Here precision is in percentage, but earlier paragraph external reproducibility is shown in real unit numbers. I suggest please be consistent throughout the methodology. It should be better to report everything in percentage, otherwise it's difficult for reader to understand the flow.

> Reply: this is a misunderstanding. We referred to the manufacturer's data sheet here, which is specified in the revised version of the paper.

Para 205 second line: Here salinity in unit but in next line it's in per mil. Be consistent. Either psu or per mil.

Reply: this is again a misunderstanding. The salinity of the seawater standards we used was measured with an Optimare Precision Salinometer that has been calibrated with IAPSO Standard Seawater. Therefore, the unit "permil" may be assigned to these values (Supplement Table S3). As this could be misleading, we have omitted the unit in the revised version of the paper.

Para 255, last sentence: This is a negative statement. Either remove this statement. If you are doubtful about the published data, don't use it.

Reply: the first author knew Professor Lutze in person. He was very accurate ,and we do not doubt his data. We simply say for clarity that the methodology was not reported. In order to be more specific, we have replaced the term "methodology of" by "the instruments used for" in the revised version of the paper.

Para 270, fourth line: I assume 1.06% of total salinity dataset. Salinity unit is also sometime expressed as %. So please clarify.

Reply: "1.06 % of the time from the total salinity record" would be correct. We have changed this in the revised version.

Para 275, first line: Rewrite! It reads 0.08% salinity lost/changed in water.

Reply: see above. "of 0.08 % of the data from the salinity record." would be correct. We have changed this in the revised version.

Para 300, first line: sometimes temp in C, sometimes in K. Be consistent. in fig 2, max water temperature shows a value less than 30. Please check the values given here or check the figure.

Reply: this is a misunderstanding, because it is common sense that temperature levels are to be reported in degree centigrade, while differences or intervals are to expressed in Kelvin. Figure 2 presents the daily mean temperatures as noted in the figure caption, while the values given in Line 296 are the extremes. We think it is self-explanatory that -0.7°C is the minimum value and 31.8°C is the maximum value.

Para 300 last line: No need to write 1sigma each time.

Reply: we respectfully disagree. Otherwise the value can be mistaken with the total data range.

Figure 4 caption: in b panel, only one station has been plotted as only one red and blue pattern. It's not clear what stations. I assume for Holtenau.

Reply: Reviewer 1 is right, the figure caption is ambiguous. It rather should read "Figure 4. Water level at Bottsand and Holtenau (a), temperature, and salinity during the Centennial Flood at Bottsand (b). The red arrows in (a) mark the salinity rises observed at Bottsand". We have changed this in the revised version of the manuscript.

Para 420 first line: low air temperature in winter.

> Reply: done.

Figure 7: in text temperature changes are discussed first, but in figure it comes in the lower panel. inconsistency!

> Reply: as a convention, we always plotted the salinities in the top panel (see Figures 2 and 3).

Section 4.2. title: In results and everywhere, title is focused on Schobüll . Now Husum Bight. Not consistent

> Reply: Schobüll is located on the north-eastern side of part of inner Husum Bight. This has been stated in Line 112 of the submitted version already.

Para 600, last line: initial hypothesis of what? Not linked properly

> Reply: the initial hypothesis of the present study has been given in Line 66of the submitted version. For clarification, we have added a sentence to the end of this paragraph of the revised version, saying that "Temperatures and salinities of intertidal waters indeed showed a different variability than those of surface waters further off shore.".

Para 605: This is a result, not conclusion of the study

> Reply: the values have been repeated here to fuel the reader's imagination. For the sake of conciseness, they have been omitted in the revised version of the manuscript.

**RC2: 'Comment on egusphere-2025-2672', Anonymous Referee #2, 23 Sep 2025**

The manuscript entitled "Hydrography of intertidal environments in Schleswig-Holstein, Germany" by Schönfeld et al. reports on a study assessing the current status of marine marginal environments using a 36-month time series of abiotic variables measured from two systems located at the Baltic Sea and the North Sea. The authors used a large temporal dataset resulting from an impressive sampling effort. The authors also reported changes in salt marsh and oyster occurrence. This is an interesting and important article. It offers valuable insights into short and long-term changes in marginal habitats. The contribution of this research to filling out gaps on the impacts of global change on fragile ecosystems is well-described and meets the criteria laid down for publication in Biogeosciences.

> Reply: we thank the Reviewer 2 for the positive review appreciate the recognition of our efforts to create the time series presented in this paper.

The manuscript is generally well-written and could be accepted after some revisions. My main concern is about the lack of information regarding the ecology of salt marshes and Pacific oyster in the Introduction.

> Reply: the ecology of the salt marshes is beyond the scope of the present study. Geochemical, microbial and faunistic data would have to be included, which are mostly not available. The floral successions have been documented in the literature much better. They are of relevance to the present study because halophytes directly respond to the hydrographical dynamics we have captured, e.g. inundation frequency. Therefore, we

have summarized the past and present state of the salt marshes in the "1.1 Geographical and environmental setting" chapter.

Responses of these to organisms to extreme events are indeed discussed. The Materials and Methods section should also provide details about the sampling strategy employed to document biotic responses (salt marsh and Pacific oyster populations) to extreme events.

Reply: there was no intention neither a sampling strategy to document biotic responses to extreme events. It would have employed a biomonitoring scheme with short sampling intervals, which is beyond the knowledge and capacities of the authors. None-the-less, observations were made by accident, in particular a temporary shell lag before the groynes off Schobüll. This observation was set into a context by exploring the hydrographical data we have obtained.

Finally, the conclusion is too long, and in some instances, this section presents information that should appear in the Results section (For instance, see lines 604-606 and lines 640-642).

Reply: done (see above).

**CC1: 'Comment on egusphere-2025-2672', Julia Lübbers, 26 Aug 2025**

I am positively surprised to see such a long and detailed hydrographic study from the North Sea and Baltic Sea coasts of Schleswig-Holstein. In the context of global warming, such baseline studies are crucial for understanding the effects and consequences of rising sea levels. Schleswig-Holstein may seem less prominent compared to major coastal areas like New York City, but this makes the availability of such a continuous record even more impressive and valuable for future research.

Reply: we sincerely thank Julia Lübbers for her comments and suggestions.

This study is of high quality, and I strongly recommend publication. I have only a few major comments and suggestions:

Photographs of the installed loggers: Including photos or schematics of the logger installations would help readers better understand how they were set up in the field.

Reply: we have taken images of the logger installations in the field and created a new Figure S1, which is provided in the revised Supplement.

Damage to the fishing rod at Schobüll: How confident are the authors that the fishing rod broke naturally rather than being damaged by human interference? Since the logger was installed near a pier, could curious passers-by have interacted with it? Would it be safer to place loggers farther away from public access to avoid possible disturbance?

Reply: human footprints indicating the access of alien violators were not recognised around the measuring point off Schobüll during our inspection visits in winter or after damage of the rods. Such footprints stay in the mud for two weeks or more. Also, the top segment of a rod broke off once in summer. The fracture was so high that no one could reach it. Consequences of installing the measuring points further away from the general public were longer approaches in difficult terrain and difficulties with levelling over long distances and unstable ground. Since these retrospective considerations did neither affect

our data logger deployment decisions nor the results of the study, we abstained from discussing the aspect of avoiding possible vandalism in the revised version of the manuscript.

Evidence for the mass oyster mortality: The manuscript mentions a "mass occurrence of empty oyster shells" at the Schobüll station. Could the authors provide quantitative estimates (e.g., number of shells per square metre) or photographs to document this observation and confirm that it was indeed a mass mortality event?

Reply: we have screened the 63-2000 µm size fraction of foraminiferal monitoring samples taken annually at the vegetation boundary off Schobüll in late October or early November. Indeed, the concentration of shell fragments in the 0-1 cm surface sediment was 53 per 10 cm$^3$ in 2023 and 304 per 10 cm$^3$ in 2024, i.e. six times higher. The values have to be taken with caution because the 2024 sample was taken more than six months later than the shell lag was observed, and the surface area was 2 x 10 cm$^2$, hence very small as compared to a human footprint. Furthermore, platy shell fragments could be washed away by wave action, covered by mud during the summer months, or blended with the underlying sediment by bioturbation. We therefore abstained from quoting these figures.

The shell fragments were identified by naked eye in the field as juvenile Pacific oysters. When we attempted to quantify their abundance, we discovered under the binocular microscope that we have mistaken barnacle plates with oyster shell fragments. In fact, they belong to *Austrominius modestus*, an invasive barnacle, which proliferates at warm temperatures and is endangered by temperatures below -5°C (O'Riordan et al., 2020). A period of strong frost between 29-Nov-2023 and 1-Dec-2023, with daily mean temperatures of up to -10.2°C, may well have caused a mass die-off, which may have produced the shell lag we have observed. The respective subchapter of the Discussion, the last paragraph of the Conclusions, and the respective sentences of the Abstract were corrected accordingly.

**Literature cited in the Response Letter:**

O'Riordan, R.M., Culloty, S.C., Mcallen, R. and Gallagher, M.C.: The biology of Austrominius modestus (Darwin) in its native and invasive range, Oceanography and Marine Biology: An Annual Review, 58, 1–78, 2020.

Schönfeld, J.: Monitoring benthic foraminiferal dynamics at Bottsand coastal lagoon (western Baltic Sea), Journal of Micropalaeontology, 37, 383–393, 2018.

---

## Author Response (AR2)

**MS No.: egusphere-2025-2672, posted as preprint on EGUsphere for public review and discussion on 7 Aug 2025, revised version uploaded on 14 November 2025.**

**Title: Hydrography of intertidal environments in Schleswig-Holstein, Germany.**

**Authors: Joachim Schönfeld (jschoenfeld@geomar.de), Hermann W. Bange, Helmke Hepach, and Svenja Reents.**

**Response to Reviewer comments**

Dear Editor,

we sincerely thank the Associate Editor Perran Cook and an anonymous Reviewer for their constructive comments and suggestions. We acknowledge that the reviewer found that the revised version of the manuscript, in particular the Methods section, is much clearer. Referee 1 noted that there is still an inconsistency in the description of the data depicted in Figure 7 and the enumeration of Figures cited in the text, and suggested an emendation for the sake of clarity.

We have addressed these issues, include a response to the comments, and provide a revised manuscript that we believe is markedly improved. In the following, the Reviewers' comments or questions on the manuscript are given in black, and our response is highlighted in blue and indented.

**Associate editor decision, Perran Cook**

Referee 1 has now reviewed the revised manuscript and is generally satisfied with the changes. They do however note there is still an inconsistency with the order of data presented in Fig 7 and the text. I suggest the authors re-order the mention of data in the text to overcome this and also refer to fig xa, xb etc for clarity as suggested by the reviewer.

> Reply: we thank the Associate Editor for his suggestion. We have addressed the issues, re-ordered the description of data in Subchapter 4.1 Seasonal dynamics of temperature and salinity in Bottsand lagoon. We also added the panel numbers to the citation of Figures in the text, which indeed provides an easier reference for the reader.

**RC1: Anonymous Referee #1, 09 Dec 2025**

The revised version of "Hydrography of intertidal environments in Schleswig-Holstein, Germany"-by Schönfeld et al., addressed most of the comments successfully. The methodologies are much clearer now. I am only concerned about one reply against Figure 7 "Reply: as a convention, we always plotted the salinities in the top panel (see Figures 2 and 3)." This can't be an explanation. The text and figures need to follow each other to make it understandable to readers. I would suggest changing it. At least, use the figure and figure panel numbers (Exam., 2a, 5b, etc) when citing in text.

> Reply: done, see above.